# Experimental Demonstration of Attoseconds-at-Harmonics at the SASE3 Undulator of the European XFEL

Andrei Trebushinin [1,*], Gianluca Geloni [1], Svitozar Serkez [1], Giuseppe Mercurio [1], Natalia Gerasimova [1], Theophilos Maltezopoulos [1], Marc Guetg [2] and Evgeny Schneidmiller [2]

1   European XFEL, Holzkoppel 4, 22869 Schenefeld, Germany
2   Deutsches Elektronen-Synchrotron, Notkestrasse 85, 22607 Hamburg, Germany
*   Correspondence: andrei.trebushinin@xfel.eu

**Abstract:** We report on observations of single spike spectra (3–13% of events) upon employing a previously proposed method for single spike generation via harmonic conversion. The method was tested at the soft X-ray SASE3 undulator of the European XFEL. The first part of the undulator allows one to amplify bunching at the fundamental as well as the higher harmonics. The downstream undulator is tuned to a harmonic, the fourth in our case, to amplify pulses with a shorter duration. We estimate the generated pulse duration within such a subset of short pulses at a level of 650 as. Considering the demonstrated probability of single spike events, this method is attractive for high repetition-rate free electron lasers.

**Keywords:** FEL; attoseconds; harmonic

## 1. Introduction

Bunching at harmonics driven by amplification of the fundamental has drawn the attention of researchers in the past and has been described in numerous publications, e.g., [1–4]. The authors of [4] proposed to use this effect for generating single spike sub-femtosecond pulses at free-electron laser facilities. The proposed setup requires minimal hardware manipulations and only the availability of variable gap undulator cells. In our contribution, we present an experimental demonstration of obtaining these nearly single-mode radiation pulses at the SASE3 undulator of the European XFEL with a two stage setup. For brevity, we shall call this the "attoseconds-at-harmonics" method.

During the amplification process in free-electron lasers (FELs), see Figure 1, the longitudinal phase space of the electron beam evolves along the undulator: in the linear regime it gradually undergoes sinusoidal energy and density modulations at the wavelength of the fundamental harmonics, as the electrons "rotate" in an FEL bucket; see the upper panels in Figure 2. At the onset of saturation, these modulations become non-linear and deviate from the sinusoidal shape. This leads to the growth of a rich harmonic content, shown in Figure 1A. The growth of bunching at harmonics is rapid and non-linear, which is characterized by a power law dependence with respect to the fundamental ($b_n(t) \sim b_1^n(t)$), see Figure 1B, where $b_n$ denotes the bunching at the corresponding $n$-th harmonic.

We illustrate the amplification process with snapshots of the electron beam and radiation properties in the bottom panels of Figure 2, which show the radiation power dependence, bunching factors, and radiation spectra. Up to the 6th undulator cell, before reaching saturation, we observe growth at the fundamental harmonic while the bunching at the 4th harmonic remains at the noise level. This corresponds to a long plateau in Figure 1B, where each mapped point corresponds to the bunching factor calculated at a given position along the electron beam. At the 8th cell, we see a rapid harmonic growth. This corresponds to the linear part of the graph on the right panel of Figure 1B, which confirms (actually, this relation $b_n(t) \sim b_1^n(t)$ strictly holds in steady state regime, while for

the time-dependent simulation results, Figure 1, the slope for the 4th harmonic deviates from 4, although this is still the power law dependence) the power law dependence of the higher harmonic bunching with respect to the fundamental. A deviation from this linear dependence indicates saturation.

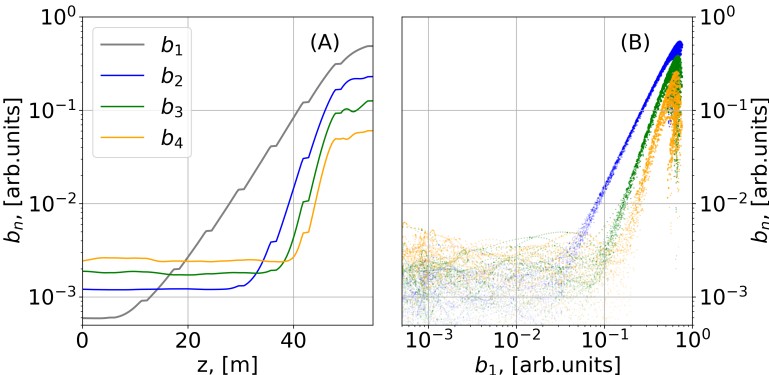

**Figure 1.** Simulation of the harmonics bunching growth with the GENESIS 1.3 code [5]. (**A**): Evolution of the bunching factor for the fundamental (1st), 2nd, 3rd, and 4th harmonic. The maximum value of bunching along electron beam is presented. (**B**) The dependence of the harmonics bunching on the bunching at the fundamental. The initially negligible level of the higher harmonics content is followed by a linear dependence with different slopes. Afterwards, this is followed by a hook-like dependence representing saturation.

    Upon reaching saturation at 42 m, shown in Figure 2, the bunching at the 4th harmonic is proportional to that at the fundamental raised to the power of 4. Afterwards, the beam radiates in the downstream radiator tuned to the given harmonic, as depicted at Figure 2d. The power law dependence leads to effective suppression of spikes with low values of bunching and to an increase in the relative height of the dominant spikes which results in the reduction of the width of the spike and total pulse duration.

    The authors of [4] proposed to use this effect for generating sub-femtosecond pulses with two or *multi*-stage undulator schemes. In the first stage, one creates bunching at the harmonic of the fundamental $\lambda$. The following stages are successively tuned to the wavelengths that correspond to the harmonics of the previous stages ($\lambda/n$). These can be sequences, for example, 1→2→4, 1→4, 1→3, etc. In this way, lasing at the harmonics starts from a substantial level of bunching created at the previous stages.

    The idea behind this scheme is to generate the events with a minimum number of spikes in the first stage and to reduce their effective number and width further by exploiting prominent bunching at the harmonics in the following stages. This way, pre- and post-spikes are greatly mitigated, as we show in Figure 2c. This allows for the generation of single spike sub-femtosecond pulses on a statistical basis, at the level of several percent of the total number of pulses [4]. This method is attractive for high repetition rate machines, since one can post-select the ensemble of events with desired pulse duration and other properties using non-invasive spectral or temporal diagnostics.

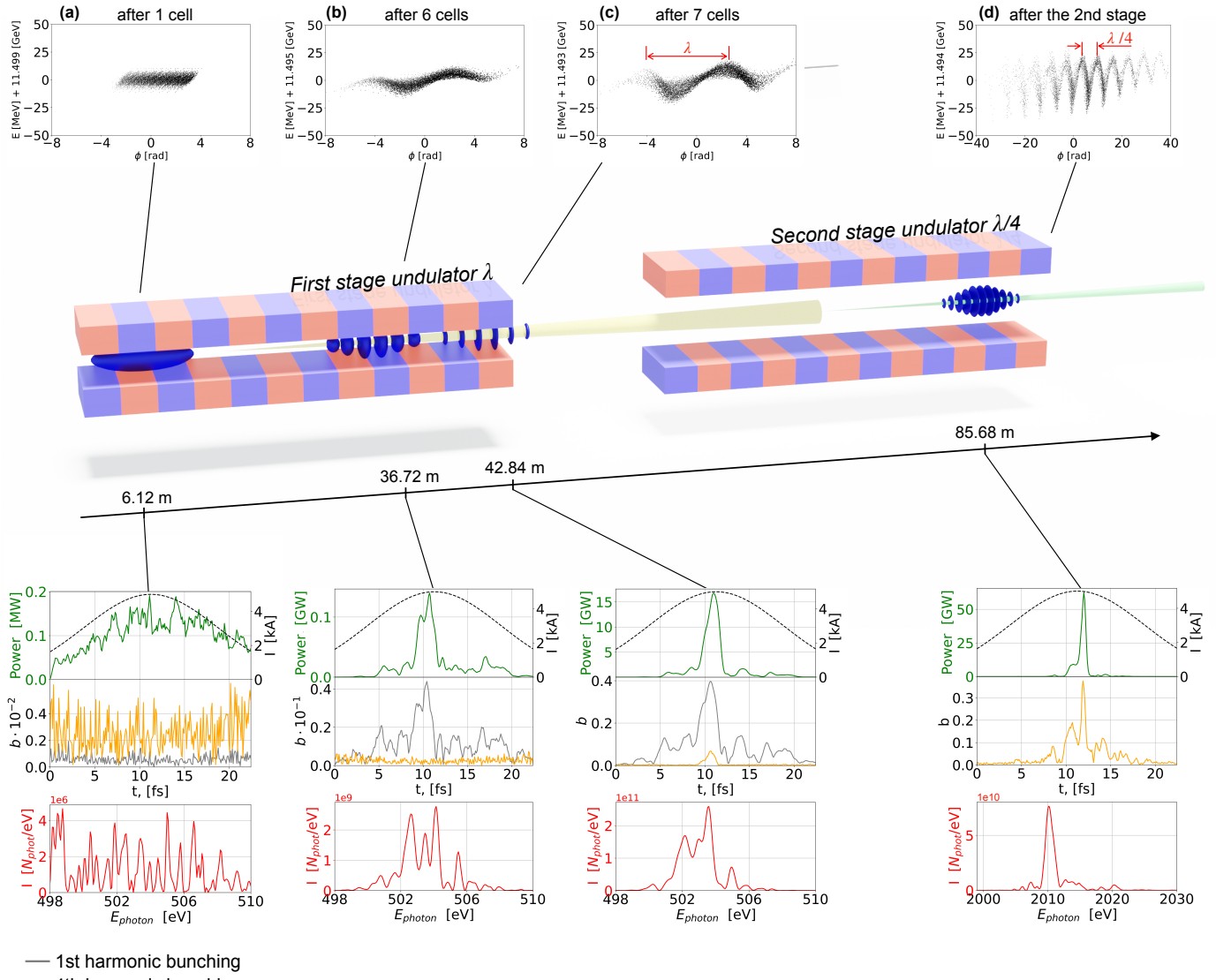

**Figure 2.** The attoseconds-at-harmonics setup, the four top and bottom sub-figures (**a**–**d**) illustrate the amplification process along the undulator. At the top, we present FEL buckets and at the bottom the radiation and electron beam parameters: power dependence in time domain (green line), bunching factors (gray and yellow lines), and the spectra (red line). (**a**–**c**) The lasing process at the first stage and (**d**) the resulting radiation and the electron beam parameters after the second stage.

## 2. Experiment at SASE3 Undulator at the European XFEL

We performed two separate experiments using the two stage setup at the SASE3 undulator of the European XFEL. The second stage was set to the 4th harmonic of the first one. In Table 1, we present the experimental parameters.

**Table 1.** Experimental parameters.

| Experiment | $E_{ebeam}$ (GeV) | $E_{1h}$ (eV) | $Cells_{1h}$ | $XGM_{1h}$ (µJ) | $E_{4h}$ (eV) | $Cells_{4h}$ |
|---|---|---|---|---|---|---|
| 675h4 (#1 and #2) | 14 | 675 | 10 | 18 | 2700 | 1 |
| 503h4 (#1 and #2) | 11.5 | 503 | 9 | 52 | 2012 | $6 \rightarrow 8$ |

During the first experiment (October 2020), we used a compressed 100 pC electron beam to minimize the amount of SASE spikes in the first stage tuned to the fundamental

photon energy of 675 eV, the spectrometer resolution was 0.13 eV [6]. The pulse energy downstream from 10 active undulator cells was at the level of 18 µJ. We recorded spectra of the fundamental, finding their relative bandwidth at a level of 0.5 %, see Figure A1a. To optimize the performance of the high harmonic bunching, we stopped FEL amplification at the fundamental at the beginning of the nonlinear regime. The second stage was set to $4 \times 675$ eV $= 2700$ eV with one undulator closed. The spectrometer resolution at this photon energy was 0.7 eV. We aimed at seeing only the effect of coherent radiation at the harmonic without further amplification. In the recorded spectra, as expected, we observed sporadic events with spikes dominant over their satellites, Figure 3a. The data from this experiment are represented by the run identifier 675h4#1 in the present article (notation like 675h4#1 encodes the main parameters of an experimental run; the first number stands for the energy at the fundamental harmonic, h4 represents which particular harmonic is considered, and #1 is the number of the run). The results were reproducible after optimization attempts, run 675h4#2.

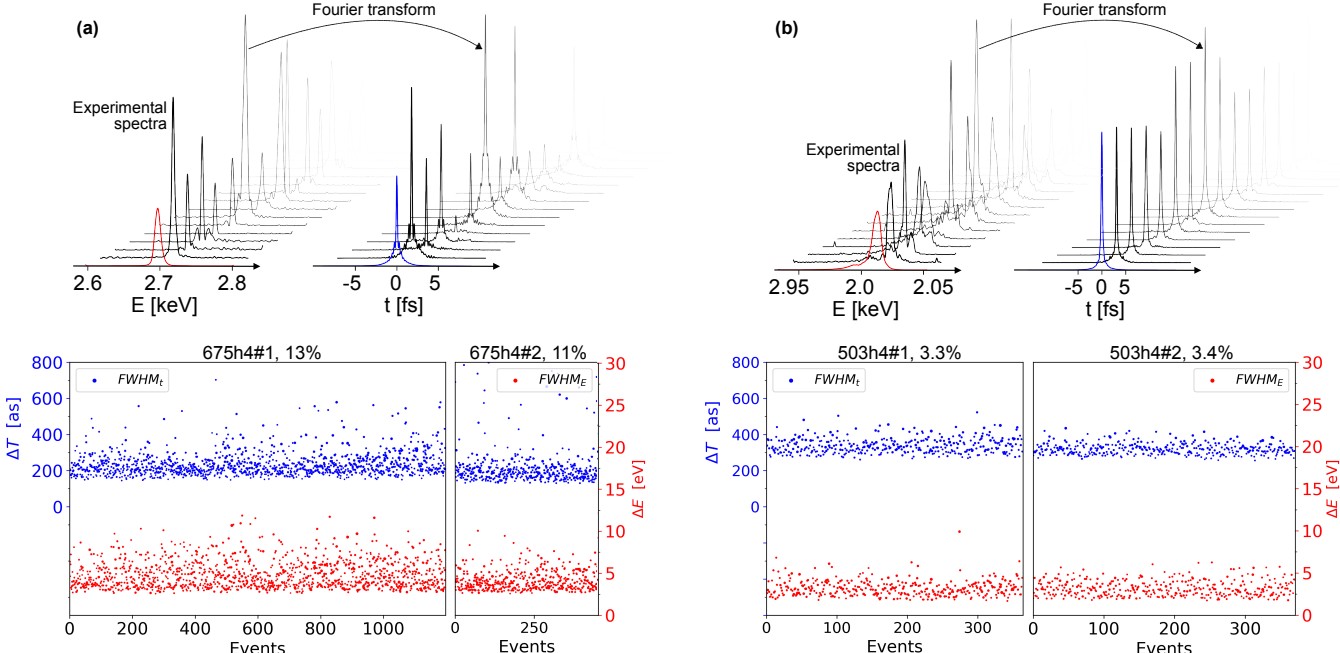

**Figure 3.** Results of data analysis. The filtered experimental data from experiment 675h1 are presented in (**a**) and those from experiment 503h1 are shown in (**b**). We present the filtered spectra in the top right panels of the sub-figures and the corresponding time domain distributions obtained by Fourier transformation in the top left panels. In the bottom panels, we show the width of the peaks: blue and red dots correspond to time and spectrum characteristics.

One of the challenges of the experiment was the diagnostics of the pulse energy. It was measured with an X-ray Gas Monitor (XGM) [7]. In two pulse operation, this device is sensitive not only to the fundamental but also at its harmonics.

During the second experiment (September 2021), we tuned nine undulator cells to the fundamental photon energy of 503 eV with a spectrometer resolution of 0.2 eV. Similarly, we optimized the electron beam compression to minimize the number of spectral spikes and acquired spectra, run 503h4#1, Figure 3b. We tuned the second stage to the fourth harmonic, and recorded radiation spectra again, with a resolution of 1.2 eV at 2012 eV. In run 503h4#2, we used *six* closed cells; after closing *two* more cells, we observed a minor increase in pulse energy from 57 µJ by 11 µJ.

*Data Analysis*

During the experiments, we collected raw spectra from the SASE3 soft X-ray spectrometer [6]. We filtered this set of spectra for single spike events only. We relied on the fact that, on a statistical basis, a single spike event in a spectrum corresponds to a single spike event in the time domain.

At first, we extracted raw data from the SASE3 spectrometer, see Figure 4a, subtracted the background, and proceeded with the application of a noise reduction algorithm, see Figure 4b. The last step was needed to facilitate the finding of peaks in the spectra, see Figure 4c. The number of single spike events was at the level of 11–13 % for the first experiment and 3% for the second experiment. The filtered events and the corresponding values for the minimal pulse duration are presented in Figure 3.

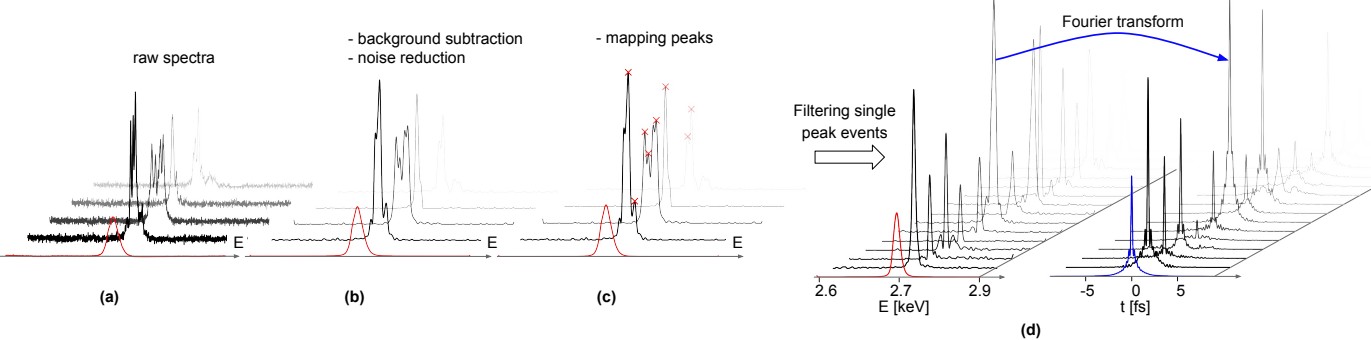

**Figure 4.** Data analysis procedure: (**a**) raw spectra extracted from the SASE3 spectrometer; (**b**) background subtracted, noise reduced; (**c**) spectrum peaks mapped; (**d**) single peak events chosen, Fourier transform performed to estimate *minimal possible* pulse duration. The red/blue line in front of the black lines represents averaged spectrum; the black line shows single shot events. Horizontal axes of (**a**–**c**) represent the energy span.

We refer to minimal pulse duration because the actual time domain power distribution is unknown in these experiments; in fact, we plan to conduct measurements with the angular streaking technique [8] to measure the actual temporal duration. At the moment, we lack direct temporal diagnostics for the attoseconds-at-harmonics experiments, and we can only provide the lower estimate of the pulse duration under the assumption of Fourier limit. We performed a Fourier transform of the spectral amplitude, in other words, we assumed a flat phase across the spectrum. Therefore, we only obtained the *minimal possible* pulse duration. Results are presented in Table 2 with the spread expressed with median absolute deviation; we also report here our estimate of the real pulse duration $(\Delta t_e)_{FWHM}$ based on the reasoning in Appendix B.

**Table 2.** Experimental results.

|  | 675h4#1 (as) | 675h4#2 (as) | 503h4#1 (as) | 503h4#2 (as) |
|---|---|---|---|---|
| $(\Delta t_{eFT})_{FWHM}$ | $234 \pm 29$ | $234 \pm 27$ | $338 \pm 27$ | $319 \pm 18$ |
| $(\Delta t_e)_{FWHM}$ |  | $650 \pm 140$ |  |  |

## 3. Discussion

As we indicated before, the attoseconds-at-harmonics method will be particularly suitable for high repetition-rate machines. For example, the European XFEL allows generating up to 27,000 photon pulses per second (2700 per train of electron bunched, trains go at 10 Hz repetition rate) [9]. Such a high repetition rate is a unique feature of superconducting FEL drivers. In this case, the attoseconds-at-harmonic method can potentially provide from 80 up to 350 single spike events per train, ten times per second.

The numbers in Table 2 provide only a rough estimation for the pulse duration. In these results, we evaluated both the minimal possible $(\Delta t_{eFT})_{FWHM}$ and the actually expected pulse duration $(\Delta t_e)_{FWHM}$. The estimations on the minimal possible pulse duration are based on a Fourier transform of the spectral amplitude and on the measure of the spread of the resulting distributions. The estimation for the actually expected pulse duration is more intricate and is explained in detail in Appendix B.

We expect to conduct further experiments: firstly, we aim to provide an absolute measurement of the harmonic pulse energy with the XGM diagnostic and, secondly, to perform angular streaking experiments to reveal the time domain of such pulses. The latter will give us a clear indication of the real pulse duration and validate the analysis of the present paper. Additionally, further theoretical analysis on statistical properties of such pulses may be possible.

Finally, we should place the attoseconds-at-harmonic method in the context of the many techniques for obtaining sub-fs level single spike pulses that have been proposed and/or tested at different facilities. A comprehensive overview of these methods is given in [10]. Given the statistical nature of the attoseconds-at-harmonics method, it should be clear that this technique might not be useful for every user's experiment, as it is based on a "post-selection" of short pulses out of an ensemble. Nevertheless, one of the benefits of the attoseconds-at-harmonics method is that it is free to implement and does not require any additional hardware to be installed, except for the availability of variable gap undulators.

## 4. Conclusions

In this paper, we show experimental results on the generation of quasi-single spike FEL pulses by applying the attoseconds-at-harmonics method. We estimate that a fraction (3–13%) of such pulses can be as short as several hundred attoseconds (650 as). We expect that this work will be followed up, in the future, by direct time-domain measurements with the angular streaking technique at the European XFEL.

**Author Contributions:** Conceptualization: authors contributed equally; Methodology, A.T., G.G., S.S., G.M. and E.S.; Software, A.T.; Validation, G.G., G.M., T.M., M.G. and E.S.; Formal analysis, A.T. and G.M.; Investigation, A.T., G.M., N.G., T.M., M.G. and E.S.; Resources, G.M., N.G. and T.M.; Data curation, N.G. and T.M.; Writing—original draft, A.T.; Writing—review & editing, authors contributed equally; Visualization, A.T. and S.S.; Supervision, G.G., S.S. and E.S.; Project administration, G.G. and E.S. All authors have read and agreed to the published version of the manuscript.

**Funding:** The cost of the article was covered by the journal using one of the author's voucher discounts (100%). This research received no extra funding.

**Institutional Review Board Statement:** Not applicable.

**Informed Consent Statement:** Not applicable.

**Data Availability Statement:** The data presented in this study are available on request from the corresponding author. The data are not publicly available due to requirements of internal authorization with the European XFEL account.

**Acknowledgments:** We thank Serguei Molodtsov for his interest in this work.

**Conflicts of Interest:** The authors declare no conflict of interest.

## Appendix A. Fundamental Harmonic in the Experiments

Before setting up lasing at the 4th harmonic we collected spectra at the fundamental from the first stage undulator and analyzed their statistical properties, which we show in Figure A1. We assume Gaussian statistics even if we are not in the linear regime to provide a crude estimate of the radiation statistical properties. Under this assumption, we can estimate its group duration (Group duration is the radiation duration at a given frequency. This should not be confused with the total pulse duration if the time-frequency chirp is present [11]. The strict mathematical definition is given in Equation (34) and Figure 10 in the referred article.).

Due to the lack of longitudinal electron beam phase space diagnostics we cannot directly measure energy chirps. However as the calculated group duration weakly depends on the photon energy across the SASE spectra, the higher orders of the electron- and photon beam energy chirp can be neglected and chirps are predominantly linear. As in both experiments the radiation spectral bandwidths are approximately 1.5 times larger than the expected SASE bandwidths, we deduce that the pulse durations is about two times larger than the average group durations, resulting in *average* duration of about 7.5 and 5 fs for first stage radiation during the experiments at 675 eV and 503 eV respectively.

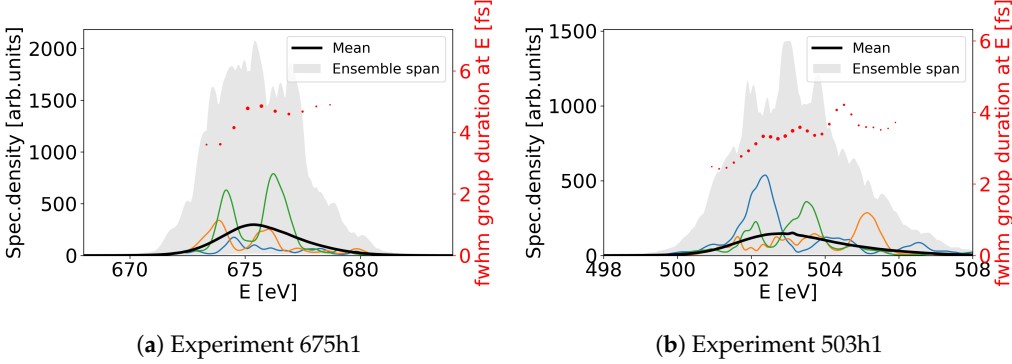

(**a**) Experiment 675h1        (**b**) Experiment 503h1

**Figure A1.** SASE spectra at the fundamental harmonic obtained during the experiments. We depict group duration at the given photon energy with red dots, the ensemble-averaged spectrum is marked with the black line and with the blue, orange and green colors we show single events. The pale grey region corresponds to a zone between maximum and minimum spectral values in the ensemble.

## Appendix B. Results of Numerical Simulations

We numerically replicated the experimental results from experiment 675h1 with the Genesis 1.3 v4 FEL simulation code [5]. In this simulation we generated 1000 events, showing good agreement with first, Figure A2a, and second stage experimental spectra, Figure A2b. We deduced the electron beam compression parameters, unknown to us, by the simulations tuned in such a way as to replicate the performance and SASE spectra from the first stage undulator, Figure A2a. The 4th harmonic bunching of the electron beam was amplified in the second stage undulator what we present in Figure A2b in comparison with experimental spectra 675h1 run.

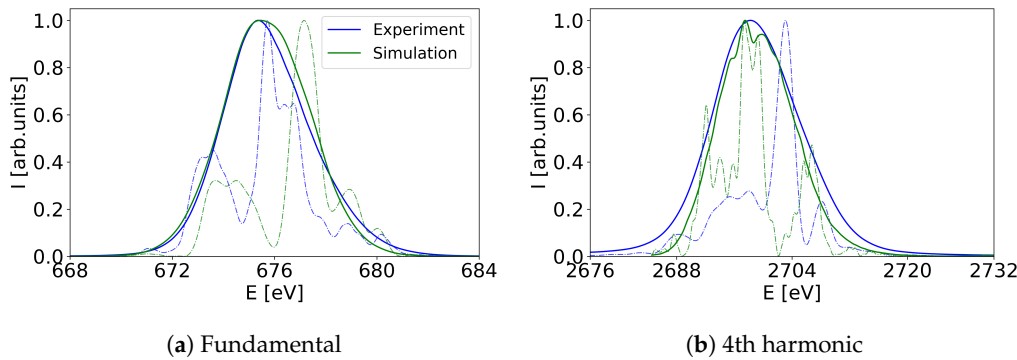

(**a**) Fundamental        (**b**) 4th harmonic

**Figure A2.** Simulation results in comparison to the experiment 675h1. Solid saturated lines denote ensemble average spectrum, dash-dotted pale lines represent single shot events.

Out of 1000 simulated events from the second undulator stage 58 contain a single pronounced spike (i.e., 5.8%) with minor satellites in the spectrum. We present a typical spectrum in Figure A3.

Simulations provide full information on both time and frequency domains of the FEL radiation and allows us to empirically find the ratio of the actual pulse duration $(\Delta t_s)_{FWHM}$

and the Fourier limited one $(\Delta t_{sFT})_{FWHM}$. The ratio is equal to $N = \left\langle \frac{(\Delta t_s)_{FWHM}}{(\Delta t_{sFT})_{FWHM}} \right\rangle = 2.8 \pm 0.5$, where $\langle \ldots \rangle$ denotes averaging over an ensemble. In other words, we expect the actual pulse in the time domain to be 2.8 times longer than the Fourier limited estimate. Applying this coefficient to our experimental results, we obtain an estimate of average pulse duration at a level of $(\Delta t_e)_{FWHM} = N \times (\Delta t_{eFT})_{FWHM} = 650 \pm 140$ as.

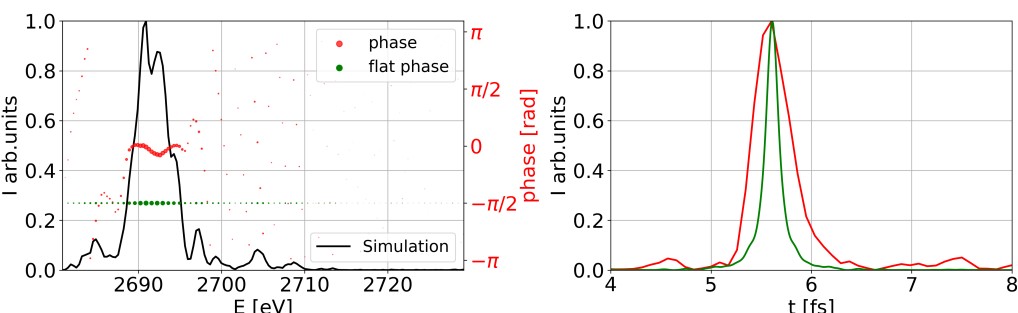

**Figure A3.** Simulated single spike event. The radiation spectrum with a single spike is presented on the left (black line). The actual phase is presented with red dots, the corresponding time distribution is presented with the red line on the right. The green dots on the left sub-figure represent the suggested assumption of the Fourier limited pulse (a flat phase across the spectrum). The Fourier limited pulse is shown by the green line on the right.

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
