# Peer review of "Experimental Demonstration of Attoseconds-at-Harmonics at the SASE3 Undulator of the European XFEL"

_photonics, doi:10.3390/photonics10020131_

Round 1

Reviewer 1 Report

The authors present the experimental results on attosecond pulse generation at the European XFEL using harmonic generation. This method is quite simple and well suited for a high repetition rate FEL. The demonstrated results are an important milestone in the development of attosecond FEL technology. Hence, I recommend the manuscript for publication.  At the same time, I suggest that the authors address the following question in a revised manuscript:

1. line 90: what is the difference between the experiments 1 and 2?
2. line 115: what is the definition of the group duration?
3. What is the spectral resolution of SXR monochromator used for measurements?

Author Response

Please, see the review in the attached file.

Reviewer 2 Report

This paper presents experimental and theoretical results of a scheme to generate sub-femtosecond X-ray pulses. This is a very important area of research and the results are encouraging. The paper presents strong evidence that the scheme is successful, although there is not yet a definitive proof that the pulses have a duration less than 1 fs. However, there are plans to perform the required measurements. The paper should be published as it represents important progress in the field.

The paper is generally well organized and presented, but there are a number of minor revisions that would improve it.

The choice of the label “attoseconds at harmonics” is confusing, especially in the title of the paper where “demonstration of attosecond at harmonics at the SASE3 undulator” looks like a typographical error and is only explained in the introduction much later. Might I suggest that the phrase be either used always in inverted commas; or written as attoseconds @ harmonics; or written with hyphens, attoseconds-at-harmonics, to avoid this confusion?

In the abstract, the phrase “short pulses at a level of 650 as” is a bit misleading as in the paper it is clearly stated that the minimum duration is estimated to be this. I suggest adding this qualifier. In the paper the simulations seem to indicate that the pulses are definitely sub fs, although no upper bound on the duration is given. Perhaps an upper bound could be estimated and included in the abstract.

I did not find a description of or reference to the photon spectrometer; this information is necessary. Resolving power? Wavelength range?

On page 1, line 30, the discussion of figure 2 begins. Figure 2 shows data after a number of cells, and also as a function of distance. Three seems to be a slight anomaly: 1 cell corresponds to 5 m, and 6 cells to 30 m. So a cell is 5 m long. 8 cells correspond to 42 m, so the last two cells are 6 m long. Is this correct?

In Figure 3, there are red spectra in front of the black spectra. Please explain what they are in the caption. “E” is presumable “energy”, and should be written as such.

Page 4, line 81, “we observed a minor increase” An increase from what value?

There are a number of typos and grammatical inaccuracies. Here are some suggested improvements.

Page 1, line 28. “properties at the bottom” -> “properties in the bottom”

In figure 1(a), the y axis scale refers to b1. What is the scaling for the other three bunching parameters?

Caption of Table 1. “Experiments parameters”-> “Experimental parameters”

Page 2, line 42 “prominence of dominant spikes for higher which” Prominence is not the right word here: do you mean importance, or effect? A word is missing after “higher”.

Figure 2(a). “after 1 cells” -> “after 1 cell”

Page 3, line 64 “downstream 10” -> “downstream from 10”

p. 4, line 90. “latest”-> “last”

p. 4, line 90. “to facilitate peak finding in the spectra,”-> to facilitate finding of peaks in the spectra,”

p. 4, line 95. Delete “while”

p. 4, line 99. “We performed Fourier transform” -> “We performed a Fourier transform”

Figure 4, caption. “675h1 experiment”->. “experiment 675h1” and similarly for all instances below, also of “503h1 experiment”

Table 2, title. “Experiment results”-> “Experimental results”

p. 5 Line 109. “with angular streaking technique” -> “with the angular streaking technique”

Appendix A, title. “Fundamental harmonic at the experiments” -> Fundamental harmonic in the experiments”

p. 6 line 114. Delete “just”

Fig. A1, caption, first line. “during October 2020 experiment.” This is the first mention of the date of the experiment: is it relevant? Maybe “during the experiments” is better.

Fig. A1. Third line of caption: “show a single events” Delete “a”

p. 6, line 116. “diagnose presence of” -> “diagnose the presence of” or perhaps “measure”

p. 6 line118. “across SASE spectra” -> “across the SASE spectra”

p. 6 line 119. “silumation” -> “simulation”

p. 6, line 129. “in such way to replicate” -> “in such a way as to replicate”

p. 6, line 130. “the dumped electron beam” I do not understand why the beam is dumped.

p. 6, line 140. “in time domain” -> “in the time domain”

p. 7, Fig. A3, caption last line. “Fourier limited pulse is presented with the green line.” -> “The Fourier limited pulse is shown by the green line.”

Author Response

(The authors gave the same response as above.)

Reviewer 3 Report

The authors of the manuscript entitled Experimental demonstration of attosecond at harmonics at the SASE3 undulator of the European XFEL, report the observation of single-spike spectra for the attosecond time scale soft X-ray generation,  which was tested in European XFEL. This is quite important for the development of the XFEL and its application in ultrafast physics. The manuscript and results are well organized. And the data should be published in the current form.

Author Response

Dear Reviewer,  

On behalf of all co-authors, I am thankful for your review.

Yours sincerely,
Andrei Trebushinin